# The Application of *Bacillus subtilis* for Adhesion Inhibition of *Pseudomonas* and Preservation of Fresh Fish

**DOI:** 10.3390/foods10123093

**Published:** 2021-12-13

**Authors:** Wen Zhang, Qiuxia Tong, Jiahong You, Xucong Lv, Zhibin Liu, Li Ni

**Affiliations:** Institute of Food Science and Technology, College of Biological Science and Technology, Fuzhou University, Fuzhou 350108, China; zhangwen@fzu.edu.cn (W.Z.); 17750231184@163.com (Q.T.); youjiahong666@163.com (J.Y.); xucong1154@fzu.edu.cn (X.L.); liuzhibin@fzu.edu.cn (Z.L.)

**Keywords:** *Bacillus subtilis*, *Pseudomonas*, adhesion inhibition, preservation

## Abstract

Inhibiting the growth of spoilage bacteria, such as *Pseudomonas* spp., is key to reducing spoilage in fish. The mucus adhesion test in vitro showed that the adhesion ability of *Bacillus subtilis* was positively correlated with its inhibition ability to *Pseudomonas* spp. In vivo experiments of tilapia showed that dietary supplementation with *B. subtilis* could reduce the adhesion and colonization of *Pseudomonas* spp. in fish intestines and flesh, as well as reduce total volatile basic nitrogen (TVB-N) production. High throughput and metabolomic analysis showed treatment with *B. subtilis*, especially C6, reduced the growth of *Pseudomonas* spp., *Aeromonas* spp., *Fusobacterium* spp., and *Enterobacterium* spp., as well as aromatic spoilage compounds associated with these bacteria, such as indole, 2,4-bis(1,1-dimethylethyl)-phenol, 3-methyl-1-butanol, phenol, and 1-octen-3-ol. Our work showed that *B. subtilis* could improve the flavor of fish by changing the intestinal flora of fish, and it shows great promise as a microecological preservative.

## 1. Introduction

Fresh fish spoil due to the action of a consortium of microorganisms. Specific spoilage organisms (SSOs) have the ability to produce metabolites that directly affect the sensory properties of the product, resulting in its rejection by consumers [1,2]. The inhibition of SSOs by applying appropriate preservation strategies can retain fish freshness and extend shelf life [3,4]. Many studies have focused on the application of biological preservatives in fish products, such as fish filets or surimi [5,6]. Most biological preservatives are metabolites of plants or microorganisms. They are usually sprayed, soaked in products, or added to surimi [7]. However, in whole fish, the fish tends to spoil from the intestines, so these methods are not applicable [8]. Therefore, new preservation methods are needed to encompass a broader range of fish products [9,10].

The predominant spoilage bacteria of chilled fish mainly include the *Pseudomonas*, *Aeromonas* spp., and *Shewanella* spp. [11,12]. As the microorganisms carried by fish come from the environment, infection is mainly through adhesion to the gastrointestinal tract and the gill mucosa of fish [13]. Cell adhesion is essential for spoilage bacteria’s infection of the host. Therefore, stopping bacterial adhesion is crucial to blocking infection [14]. Probiotics have been used to block pathogen adhesion. For example, *Lactobacillus* spp. with a high adhesive ability was found to inhibit the adhesion of *salmonella* spp. to epithelial cells by 62.58% [15]. In an adhesion antagonism experiment, the co-incubation of *Escherichia coli* infected Caco-2 cells with *B. subtilis* reduced the adhesion of *E. coli* strain in the cells [16]. *B. subtilis* can reduce its adhesion, colonization, and overgrowth in human intestines by inhibiting *Staphylococcus aureus* quorum sensing [17]. *B. subtilis* is currently the most commonly used probiotic in aquaculture. It can be colonized in the body by adhering to epithelial cells and regulating the composition of animal gastrointestinal flora to prevent pathogenic microorganisms from infecting the host [18,19].

Our previous study found that *B. subtilis* could inhibit spoilage bacteria and prolong the shelf life of chilled large yellow croaker (total number of bacteria, TVB-N and K value were used as indexes) [20]. Therefore, in this work, we screened six *B. subtilis* strains isolated from tilapia for in vitro and in vivo adhesion and spoilage tests. In vitro and in vivo studies were used to measure the adhesion of the *B. Subtilis* and inhibition on the adhesion of spoilage bacteria, as well as to explore *B. subtilis*’ ability to maintain the balance of gastrointestinal flora and inhibit the production of putrefactive volatile flavor substances in fish. This study may provide theoretical guidance for the development of microecological preservative preparations.

## 2. Materials and Methods

### 2.1. Experimental Materials and Strains

*B. subtilis* and *Pseudomonas* spp. (L) were isolated and identified from tilapia by the Institute of Food Science and Technology of Fuzhou University. In this study, six strains of *B. subtilis* with different adhesion properties were screened according to adhesion index (self-coacervation rate, hydrophobicity, biofilm, etc., data were not shown) for in vitro and in vivo tests. The sequence accession numbers of the strains used in this study are shown in Table 1. The selected phylogenetic tree of *B. subtilis* is shown in appendix Appendix A. The fresh tilapia was obtained from the Freshwater Fisheries Research Institute of Fujian.

### 2.2. Determination of the Adhesion Ability of B. subtilis In Vitro

Larsen’s method was employed for collecting tilapia intestinal mucus [21]. To each hole of a 96-well plate, 150 μL mucus was added and incubated at 4 °C overnight (fixed for 18 h). The residual mucus was washed twice with 200 μL sterile PBS solution. To the wells, 150 μL *B. subtilis* solution labeled with FITC was added and incubated at 30 °C for 90 min. The wells were washed twice with sterile normal saline to remove the bacteria that did not adhere, and 150 μL SDS (1%) solution was added to each well. The well plate was incubated in an oven at 60 °C for 1 h. The detection of fluorescence was carried out using a multiscan fluorometer (SpectraMax i3+MiniMax, Molecular Devices, Sunnyvale, CA, USA) with three wells in each group. SDS (1%) was used as a blank control group. The instrument settings were as follows: excitation wavelength: 495 nm; emission: 525 nm; optical position: bottom; sensitivity: auto; and replicates per sample: 12. The adhesion rate was calculated using Equation (1).
(1)$$Adhesion\;rate\;\left(\%\right)=\frac{A_{1}-A_{0}}{A_{2}}\times 100\%$$
where *A*_1_ and *A*_2_ are the fluorescence intensities of *B. subtilis* after adhesion and pure *B. subtilis* suspension, respectively; *A*_0_ is the fluorescence intensity of the blank group.

### 2.3. Determination of the Adhesion Inhibition Ability of B. subtilis to Pseudomonas spp.

According to Forestier’s method, the simulated fish intestine mucus model was prepared [22]. For competition inhibition, the FITC-labeled *Pseudomonas* spp. The L. suspension was mixed with *B. subtilis* in equal volumes. To the pore plate coated with mucus, 200 μL of the mixture was added and incubated at 30 °C for 2 h. For displacement inhibition, FITC-labeled *Pseudomonas* spp. L. suspension was added and incubated for 2 h. Non-adherent bacteria were removed, and then *B. subtilis* suspension was added, followed by incubation for 2 h. For exclusion inhibition, *B. subtilis* suspension was added first, and then FITC-labeled *Pseudomonas* spp. L. was added. After the incubation of three kinds of adhesion inhibition reactions, the cells were flushed twice with sterile PBS. The adhered cells were released and lysed with 150 μL of a SDS (1%) solution at 60 °C for 1 h. The fluorescence intensity was measured using the multiscan fluorometer in the same way as in Section 2.2. The adhesion inhibition rate was calculated using Equation (2).
(2)$$Adhesion\;inhibition\;rate\;\left(\%\right)=1-\frac{A_{2}}{A_{1}}\times 100\%$$
where *A*_2_ and *A*_1_ are the fluorescence intensity of *Pseudomonas* spp. L. adhering to the fish intestinal mucus in the presence and absence of *B. subtilis*, respectively.

### 2.4. In Vivo Adhesion Test

#### 2.4.1. Feeding Experiment of Tilapia

The suspension of *B. subtilis* was sprayed on the surface of the basal fish feed, forming a probiotic supplemented fish feed with *B. subtilis*. The content of *B. subtilis* in the feed was 1 × 10^8^ CFU/g. About 160 juvenile tilapias were randomly divided into four groups. Five milliliters of 1 × 10^8^ CFU/mL of *Pseudomonas* spp. L. suspension was added to water. The control group (L group) was fed with basic feed, and the experimental groups (C6 + L group, C15 + L group, B18 + L group) were fed with the *B. subtilis*-supplemented feed. The fish in the same group were kept in two separated plastic tanks (20 fish per tank, capacity: 50 L). Feed was supplied at ten o’clock every day. The cultivation water with a temperature of 23 °C to 25 °C. During the experiment, approximately 50% of cultivation water was changed daily, along with the purge of the unconsumed feed and fish feces. After one week of adaptation with basal feed, the feeding experiment lasted one week. The feed used in the feeding experiment was the basic feed provided by Freshwater Fisheries Research Institute of Fujian, and the main ingredients include soybean cake, bran, fishmeal, vitamins, inorganic salts, and fish oil.

#### 2.4.2. Sample Collection and Processing

After the feeding experiment, the fish were starved for 24 h. All fish were caught, half of them were slaughtered, and the other half were stored at 4 °C immediately, because it was found from the preliminary experiment results that the end point of freshness of fish stored at 4 °C was on the eighth day (TVB-N > 30 mg/100 g). Thus, at storage for eight days, the other half of the fish were also slaughtered. Five tilapias were randomly selected from each group, and their intestines and flesh were taken. Sterile water was used to wash the intestines and remove fat. About 0.200 g fish intestines and 0.500 g dorsal fish flesh were placed in 2 mL sterile centrifuge tubes, respectively, to extract DNA. The remaining fish were sliced, homogenized, and stored in sterile bags at 4 °C for the TVB-N test.

### 2.5. Determination of Total Volatile Basic Nitrogen (TVB-N)

For the determination of TVB-N, the method (GB 5009.228-2016, Beijing, China) was used [23]. The homogenized fish sample was weighed to 10.00 g and transferred to a distillation tube. To the tube, 75 mL distilled water was added. The mixture was shaken well and soaked for 30 min. The distillation process was carried out in a digiprep total kjeldahl nitrogen systems (K9840, Hanon Technologies Inc., Jinan, China), the 1.00 g MgO and 30 mL boric acid were added, and reacted for 3 min. After distillation, titrate receiving solution with 0.0100 mol/L hydrochloric acid standard solution was added, and the end color of the mixed indicator solution was purple red. Record the volume value of hydrochloric acid consumed. TVB-N content was calculated using Equation (3).
(3)$$TVB-N\;content\;\left(\frac{mg}{100g}\right)=\left(V_{1}-V_{2}\right)\times c\times \frac{14}{m}\times 100$$
where *V*_1_ and *V*_2_ are the volume of hydrochloric acid consumed by the experimental group and reagent blank, respectively (mL); *c* is the concentration of hydrochloric acid standard solution (mol/L); and *m* is the mass of the sample (g).

### 2.6. Determination of Volatile Flavor Compounds by GC-MS

The method published by Wang XF was used to determine volatile flavor compounds, with appropriate modifications [24]. Into a 15 mL extraction bottle, 2.00 g flesh were weighed. Five milliliters of saturated NaCl solution were added, and the mixture was stirred by magnetic stir bar. To the extraction bottle, 200 μL of the original concentration of 2 mg/L standard substance 2,4,6-trimethylpyridine was added. The bottle was capped, the extraction needle was inserted into it, and it was incubated in a 60 °C water bath for 5 min. The fiber head of the extraction needle was inserted into the bottle for headspace adsorption and extracted for 30 min. After the extraction, the extraction needle was inserted into the GC-MS (7890-B/5977A, Agilent Technologies Inc., Palo Alto, CA, USA) injection port, and it was removed after desorption for 3 min. The GC-MS program consisted of 40 °C (3 min), the temperature was raised to 120 °C (3 min) at 5 °C/min, and then raised to 230 °C (5 min) at 20 °C/min. The concentration of compounds was calculated according to the proportion of the peak area of flavor compounds (Equation (4)).
(4)$$\mathrm{Concentration}\;\mathrm{of}\;\mathrm{volatile}\;\mathrm{components}\;\;\left(\mathrm{mg}/L\right)=\left(A_{i}/A\right)\times C$$
where *A_i_* and *A* are the peak areas of volatile component *i* and the internal standard, respectively, and *C* is the internal standard concentration (mg/L).

### 2.7. Analysis of Changes of Intestinal Microflora in Tilapia by High-Throughput Sequencing

The DNA of fish intestine was extracted using a fecal genomic DNA Extraction Kit (Tiangen, Beijing, China). Bacterial primers 341-F (50-CCT AYG GGR BGC ASC AG-30) and 806-R (50-GGA CTA CNN GGG TAT CTA AT-30) were used to amplify the V3-V4 region of bacterial 16S rRNA genes. The sequencing library of bacterial 16S rRNA genes was generated for high-throughput sequencing, employing the TruSeqfi DNA PCR-Free Sample Preparation Kit (Illumina, San Diego, CA, USA). Next, the library was sequenced on an Illumina HiSeq2500 platform manufactured by Novogene Bioinformatics Technology Co., Ltd. (Beijing, China). The NCBI sequence read archive was used for data analysis.

### 2.8. Detection of Specific Microflora in Intestine and Flesh of Tilapia by Real-Time Quantitative PCR (qPCR)

The total DNA of intestinal and flesh microorganisms was extracted using a fecal genomic DNA Extraction Kit and a bacterial genomic DNA extraction kit (Tiangen, Beijing, China), respectively. The total DNA of intestine and flesh was detected using the SYBR Green dye method. The specific primer sequence is shown in Table 2, and it was synthesized by Shanghai Shenggong Co., Ltd. (Shanghai, China). The specific primers were verified, and the logarithm standard curve of the bacterial concentration was drawn. The PCR sequence was as follows: initial denaturation 95 °C 30 s; denaturation 95 °C 5 s; anneal 60 °C 30 s; extend 72 °C 30 s. A total of 42 cycles were performed. The dissolution curve analysis program was as follows: 95 °C for 15 s; 60 °C for 30 s; 3 °C/min to 95 °C for 15 s.

### 2.9. Statistical Analysis

The results of data processing are expressed as mean ± standard deviation. The data were plotted by GraphPad Prism 8.0 and analyzed by SPSS 22.0’s Duncan test.

## 3. Results

### 3.1. Adhesion Ability of B. subtilis In Vitro and Adhesion Inhibition to Pseudomonas spp.

As shown in Figure 1, the in vitro adhesion rate of the six strains of *B. subtilis* was in the range of 1–10%. The adhesion ability of B. subtilis C6 and B08 was the highest, followed by C15, B15 and B18, and B02. Therefore, *B. subtilis* with high (C6, B08), medium (C15, B15), and low (B02, B18) adhesion properties were selected for subsequent inhibition on the adhesion of Pseudomonas spp. and in vivo test.

Probiotics antagonize spoilage bacteria adhesion mainly through competition, displacement, and exclusion [25]. The adhesion antagonism experiment was carried out by a mucus model in vitro. In the adhesion antagonistic test, high adhesion B. subtilis C6 and B08 had the strongest inhibitory effect on Pseudomonas spp. adhesion (*p* < 0.05), while low adhesion B. subtilis B02 and B18 had weak inhibitory effects (Figure 2). The results indicate that the adhesion ability of B. subtilis is positively correlated with its adhesion antagonism to spoilage bacteria.

### 3.2. In Vivo Adhesion of B. subtilis and the Preservation Effect on Tilapia

#### 3.2.1. Effects of B. subtilis on Intestinal Flora Diversity of Tilapia

*B. subtilis* C6, C15, and B18 with high, medium, and low adhesion and inhibition ability, respectively, were added to the tilapia diet for an in vivo adhesion test. The structure of intestinal flora in high-throughput sequencing results of different treatment groups was analyzed by principal component analysis (PCA).

The spoilage bacteria in fresh fish mainly come from microorganisms in the water environment, most of which are aerobic bacteria [26]. As shown in Figure 3, the bacteria in tilapia intestines during storage for zero days mainly include *Fusobacteriaceae*, *Vibrionaceae*, *Flavobacteriaceae*, *Peptostreptococcaceae*, *Bacteroidaceae*, and *Pseudomonadaceae*. *Pseudomonas* spp., *Flavobacterium* spp., *Bacteroides* spp., *Vibrio* spp., *Fusobacterium* spp., and *Streptococcus* spp. are common spoilage bacteria, which easily attach to the skin, gills, and intestines of fish [27]. Conversely, the diversity of intestinal flora in the supplement *B. subtilis* C6 feed group was significantly different from that of other treatment groups (Figure 3). The relative abundance of *Moraxellaceae* and *Bradyrhizobiaceae* increased, which could provide a hypoxic environment and inhibit the growth of spoilage bacteria [28].

After eight days of cold storage, the dominant bacteria mainly included *Enterobacteriaceae*, *Aeromonadaceae* and *Streptococcaceae* (Figure 3), which are the dominant spoilage bacteria in freshwater fish [29]. As shown in Figure 4, compared with the control group, the relative abundance of *Moraxellaceae* and *Clostridiaceae* in the supplemented feed group increased at zero days of storage, while the relative abundance of *Fusobacteria* spp. and *Aeromonas* spp. decreased. Similarly, the relative abundance of *Fusobacteria* spp. and *Aeromonas* spp. in the supplemented feed group decreased after eight days of storage (Figure 4).

Therefore, *B. subtilis* plays an important role in inhibiting the initial adhesion, growth, and reproduction of spoilage bacteria, as well as in maintaining the balance of intestinal flora in fish [30].

#### 3.2.2. Analysis of Target Flora in Fish Intestines and Flesh by qPCR

The adhesion and colonization of *B. subtilis* and *Pseudomonas* spp. in fish intestines and flesh (stored at 4 °C for zero and eight days) were analyzed by qPCR. As shown in Figure 5, the growth of high-adhesion *B. subtilis* C6 in fish intestine and flesh was the highest (Figure 5a,b, *p* < 0.05). However, the growth of *Pseudomonas* spp. in fish intestines and proliferation to the flesh were significantly inhibited by *B. subtilis* C6 (Figure 5c,d, *p* < 0.05), and the total number of bacteria was the lowest (Figure 5e,f).

The results show that the *B. subtilis* C6 with high adhesion ability had obvious advantages in fish intestines and flesh, and the growth and proliferation of *Pseudomonas* spp. were inhibited.

### 3.3. Analysis of Volatile Flavor Compounds in Fish Flesh

Seventy-six kinds of flavor compounds were detected by GC-MS in different treatment groups after storage for zero and eight days (Appendix A). Based on PLS-DA model analysis, there were 35 kinds of volatile flavor compounds (variable important in projection (VIP) > 1) in each group (as shown in Appendix A). The results of principal component analysis (PCA) and difference analysis (control group and high adhesion *B. subtilis* C6 experimental group) are shown in Figure 6 and Figure 7. The samples refrigerated at 4 °C for zero days were located on the positive axis of PC1 (Figure 6), and the high-adhesion *B. subtilis* treatment group (H_0) were significantly different from the others. The differential substances in the H_0 group mainly included 1,3-bis(1,1-dimethylethyl)-benzene, methoxy-phenyl-oxime, 6-octadecenoic acid, and 1,3-trans,5-cis-octariene, while the Con_0 group included ethanol, alkanes, hexadecanoic acid, ethyl ester, hexadecanal, tetradecanoic acid, and ethyl ester (Figure 7a). Saturated hydrocarbons, such as tetradecane and heptadecane, have mild odors [31]; ethanol has a slight pungent taste [32]; hexadecanoic acid, ethyl ester, and hexadecanal are the main sources of fresh flavor in aquatic animals; tetradecanoic acid and ethyl ester have a slight sweet and waxy smell; and methoxy-phenyl-oxime has a fishy smell [5,33]. These are the common flavor compounds of fresh aquatic fish.

Samples refrigerated for eight days were located on the negative axis of PC1 (Figure 6), and the *B. subtilis* treatment group was significantly different from the control group. Phenol, 3-methyl-1-butanol, hexanedioic acid, dimethyl ester, octanal, heptanal, indole, 1,3-bis(1,1-dimethylethyl)-benzene, hexadecanoic acid, methyl ester, 2,4-bis(1,1-dimethylethyl)-phenol, 1-octen-3-ol, nonanal, and decanal were the characteristic flavor substances in the control group after eight days of storage (Figure 6 and Figure 7b). Indole has a strong fecal odor [34]; phenol and 2,4-bis(1,1-dimethylethyl)-phenol have the plastic rubber odor, and are the main source of putrid fish odor [35]. C4–C6 alcohols, such as 3-methyl-1-butanol, have a similar anesthetic odor [36]. Hexadecanoic acid and methyl ester have an oily waxy odor, which is also an important component of rancidity. Nonanal, octanal, heptanal, decanal, and other straight chain aldehydes have a citrus peel flavor, fresh green herb flavor, raw potato flavor, and oily nut flavor, respectively. However, when their concentration exceeds a certain threshold, they produce undesirable odors [34]. The production of 1-octen-3-ol is mainly due to fish spoilage [37].

In conclusion, the spoilage bacteria will produce phenol, ethanol, 3-methyl-1-butanol, 2,4-bis(1,1-dimethylethyl)-phenol, indole, and other spoilage flavor compounds. *B. subtilis* treatment can effectively reduce the production of these spoilage flavor compounds and maintain the flavor of fish. Moreover, the effect of high-adhesion *B. subtilis* C6 was the most significant.

### 3.4. Analysis of TVB-N Content in Fish Flesh

TVB-N can reflect the degradation of protein in aquatic products, as well as the production of non-protein nitrogen compounds and volatile nitrogen during microbial metabolism, which is one of the key factors used to evaluate the spoilage of aquatic products [38]. In this study, TVB-N content was used as the spoilage index of tilapia, and the changes in TVB-N content in tilapia flesh during storage for zero and eight days were determined. As shown in Figure 8, there was no significant difference in TVB-N content between different treatment groups after storage for day zero (*p* > 0.05). After storage for eight days, the three *B. subtilis* strains significantly inhibited the decay process, and the TVB-N content of high-adhesion C6 + L and C15 + L groups was still within the freshness range (TVB-N < 30 mg/100 g).

### 3.5. Correlation Analysis between Intestinal Flora and Volatile Flavor Compounds

Based on Pearson correlation analysis, the correlation between the dominant flora and the different volatile flavor compounds was analyzed. As shown in Figure 9, in the correlation network analysis diagram, the dominant flora was related to the different volatile flavor compounds. Indole, 1,3-bis(1,1-dimethylethyl)-benzene, 1-octen-3-ol and 2,4-bis(1,1-dimethylethyl)-phenol were positively correlated with *Aeromonadaceae*, *Fusobacteriaceae* and *Enterobacteriaceae*. However, the compounds contributing fresh and fishy flavors, such as ethanol, tetradecane, pentadecane, 2,6,10,14-tetramethyl-pentadecane, hexadecanoic acid, ethyl ester, and 1,3-bis(1,1-dimethylethyl)-benzene, were negatively correlated with *Enterobacteriaceae* and *Aeromonadaceae* and positively correlated with *Bacteroidaceae* and *Streptococcaceae*.

Therefore, *B. subtilis* can inhibit the growth of *Aeromonas* spp., *Fusobacterium* spp., and *Enterobacter* spp. during the storage of fish by regulating the intestinal flora and reducing the production of characteristic spoilage compounds, such as indole, 1,3-bis(1,1-dimethylethyl)-benzene, 2,4-bis(1,1-dimethylethyl)-phenol, and 1-octen-3-ol.

## 4. Discussion

Probiotics can antagonize spoilage bacteria by producing antibacterial substances and inhibiting adhesion. However, the prerequisite for probiotics to play a probiotic role is adhesion and colonization on the host surface [39]. In this study, high-adhesion *B. subtilis* C6 and B08 had stronger adhesion antagonistic effects on *Pseudomonas* spp. High-adhesion probiotics can effectively adhere to the gastrointestinal mucosa, exerting a space-occupying colonization effect. As a result, spoilage bacteria are hindered from contacting the intestinal mucosa, thus reducing adhesion, colonization, and reproduction. It has been found that *Lactobacillus* spp. antagonized *Listeria monocytogenes*, *Salmonella typhi*, and *E. coli* O157: H7 mainly by competition and exclusion adhesion inhibition. The *Lactobacil* lus spp. interfered with the interaction between pathogens and host cells, preventing their adhesion and further colonization [40]. Some studies showed *Lactobacillus plantarum* mainly antagonized the colonization of *E. coli* by exclusion and competition adhesion inhibition [19]. This study indicated that high-adhesion *B. subtilis* C6 had strong inhibition abilities in competition, displacement, and exclusion. The antagonistic effect of different probiotics on the adhesion of spoilage bacteria is related to many factors, such as cell surface adhesion, adhesion sites, surface proteins, and strain specificity [41].

Real-time quantitative PCR (qPCR) results showed that the in vitro adhesion model could predict the adhesion of *B. subtilis* to tilapia intestine and the inhibition of adhesion to *Pseudomonas* spp. The inhibition of high-adhesion *B. subtilis* effectively reduced the total number of bacteria and the content of TVB-N in fish intestines and flesh. *B. subtilis* is widely used to regulate the intestinal flora of fish. Previous studies have shown that *B. subtilis* can regulate the intestinal microbiota of grass carp and promote a healthy intestinal system [42]. In some studies, it was used in the preservation of fish filets because of its antibacterial effect or quorum-quenching mechanism. For example, researchers fed bass *B. subtilis* BA37, which significantly improved the survival rate and shelf life of young fish [43]. This study uses the adhesion inhibition of *B. subtilis* to *Pseudomonas* spp. to start with the intestinal tract of chilled fish, which represents a new strategy of preservation.

More interestingly, the changes in flora were measured by high-throughput sequencing, and the changes in volatile flavor substances were analyzed by GC-MS. *B. subtilis* not only maintained the freshness of fish (TVB-N and total number of bacteria) but also improved the flavor of fish from the perspective of regulating intestinal flora. In the feeding experiment of tilapia, when *Pseudomonas* spp. L. was added to the water, the main products were indole, phenol, 3-methyl-1-butanol, 1,3-bis(1,1-dimethylethyl)-benzene and 2,4-bis(1,1-dimethylethyl)-phenol, which are typical volatile components related to the putrefaction of freshwater fish [44]. However, after feeding them the *B. subtilis* diet, the variety and quantity of these compounds decreased.

At the same time, high-throughput sequencing and principal component analysis (PCA) identified *Aeromonadaceae*, *Streptococcaceae*, *Enterobacteriaceae* and *Pseudomonadaceae* as having the highest abundances in fish. The spoilage activity of *Pseudomonas* spp. is mainly due to the production of some volatile organic compounds during low-temperature storage [45]. *Pseudomonas* spp. is considered an important contributor to the production of aldehydes, alcohols, and esters. Its characteristic odor after infecting fish is best described as acidic. *Aeromonas* spp. are specific spoilage bacteria in aquatic products, and they can often be isolated from deteriorated seafood. *Aeromonas* spp. can use glucose to produce acid and gas, thus forming a disgusting smell of corruption [46]. *Fusobacteriaceae* are the main spoilage bacteria in aquatic products captured in polluted water. Their metabolites in carbohydrate fermentation pathways are mixed organic acids or alcohols [47]. The *Streptococcaceae* bacteria are common in the intestinal tract, most of which are not pathogenic [48]. However, the abundance of these bacteria decreased in the samples treated with *B. subtilis*. It is widely accepted that *B. subtilis* can regulate the intestinal flora of fish. Studies have found that the intestinal microbial structure improved after treatment, and the abundance of *E. coli*, *Shigella* spp. and *Enterococcus* spp. decreased significantly in the group fed with *B. subtilis* [7]. In general, *B. subtilis* can improve the pathogenic bacteria in this work, such as *E. coli*, *Enterococcus* spp., *Aeromonas hydrophila* and so on. *B. subtilis* can also inhibit the growth of spoilage bacteria.

## 5. Conclusions

In conclusion, we developed a new preservation method for fish products. The mucus adhesion model was used to screen for *B. subtilis* with strong inhibitory effects against *Pseudomonas* spp. adhesion. Subsequently, we conducted a tilapia feeding test. Through in vitro and in vivo experiments, the antagonistic effect of *B. subtilis* on *Pseudomonas* spp. increased with the enhancement of its adhesion ability. High adhesion *B. subtilis* C6 inhibited the colonization of *Pseudomonas* spp. and maintained the balance of intestinal flora, thus effectively reducing the formation of putrid odor compounds. Thus, *B. subtilis* is an attractive microecological preservative, and its adhesion ability is an important index for screening effective strains.

## Figures and Tables

**Figure 1 foods-10-03093-f001:**
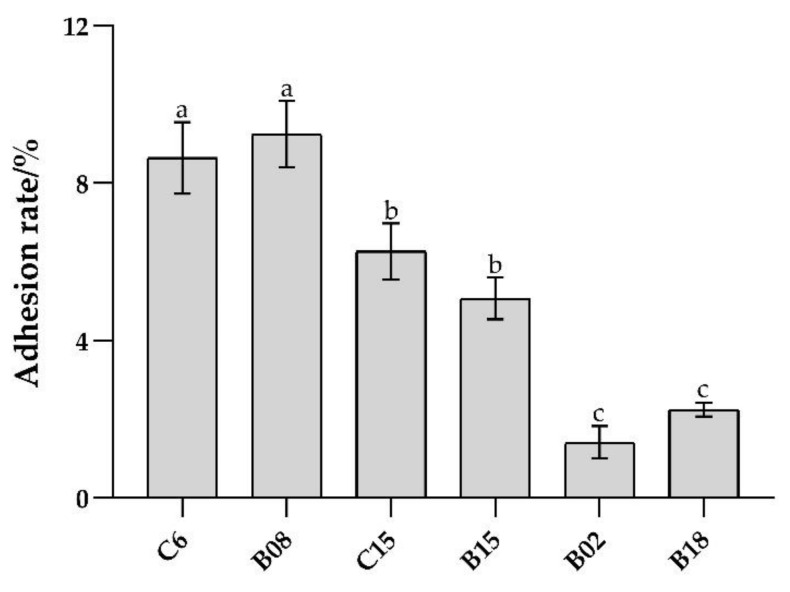
Adhesion ability of six strains of *B. subtilis* in vitro. Different lowercase letters indicate significant differences at 0.05 level (*p* < 0.05).

**Figure 2 foods-10-03093-f002:**
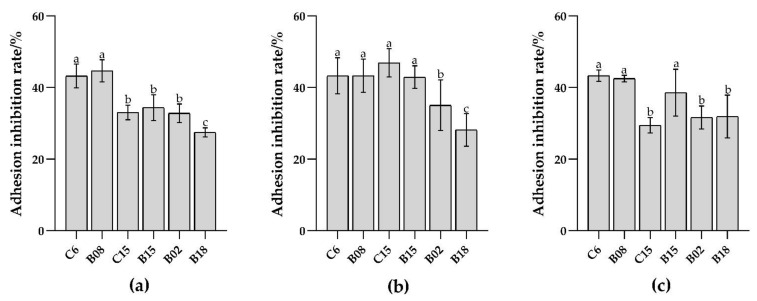
Adhesion inhibition of six strains of *B. subtilis* to *Pseudomonas* spp. L. (**a**) Competition inhibition; (**b**) Displacement inhibition; (**c**) Exclusion inhibition. Different lowercase letters indicate significant differences at 0.05 level (*p* < 0.05).

**Figure 3 foods-10-03093-f003:**
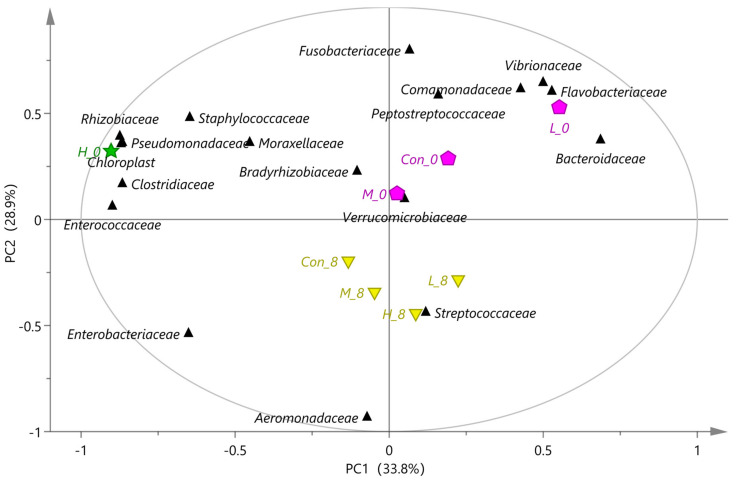
The PCA biplot plot of high throughput sequencing results of intestinal flora. The control group (Con_0: Refrigerate for zero days; Con_8: Refrigerate for eight days) was fed with common feed, and the experimental groups (C6 + L (H_0, H_8), C15 + L (M_0, M_8), B18 + L (L_0, L_8)) were fed with the supplemented *B. subtilis* feed. Different colors represent different clustering groups.

**Figure 4 foods-10-03093-f004:**
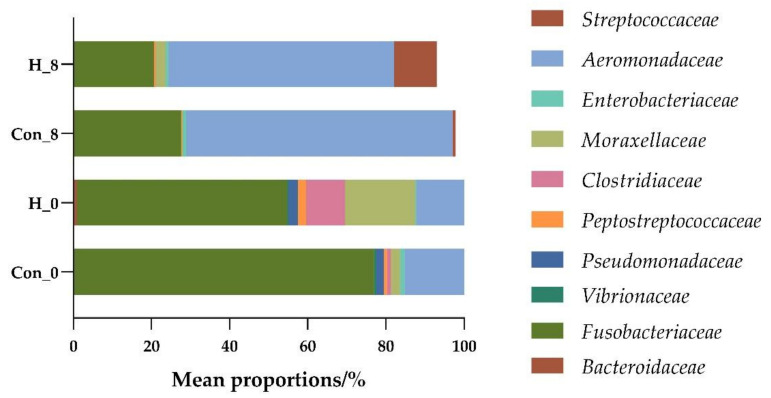
Difference analysis of the mean proportion of the top 10 flora in the high-throughput sequencing results of tilapia intestinal flora in the control group (Con_0: Refrigerate for zero days; Con_8: Refrigerate for eight days) and high adhesion *B. subtilis* C6 experimental group (H_0, H_8).

**Figure 5 foods-10-03093-f005:**
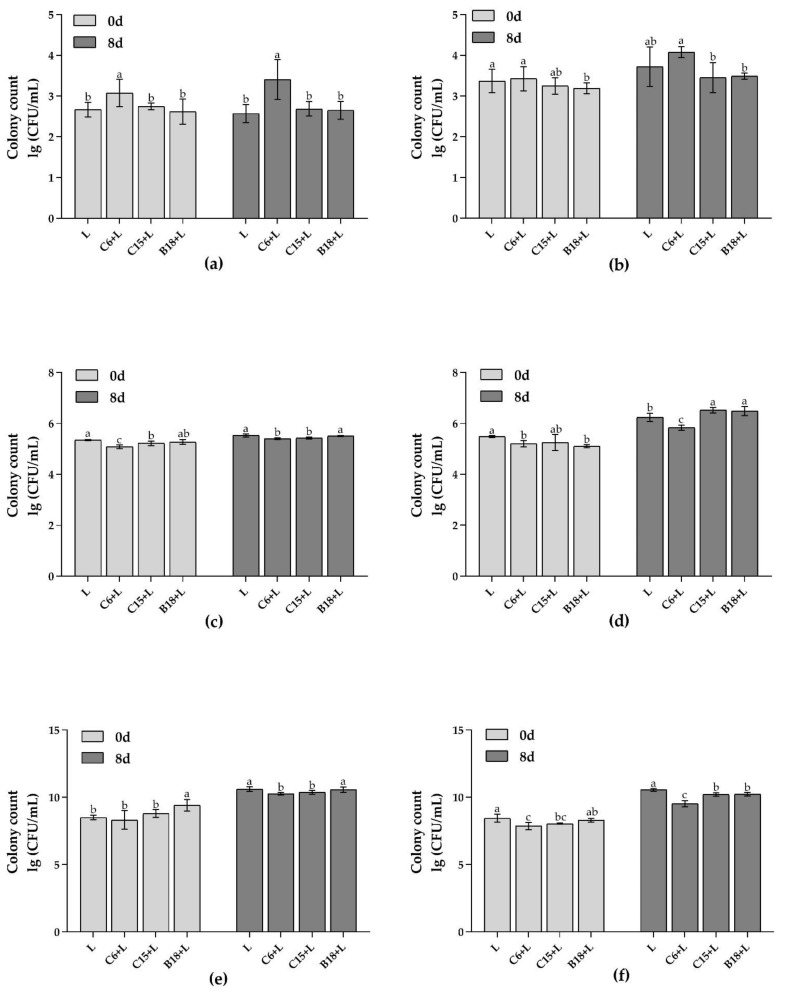
qPCR analysis of f bacterial growth in the intestine (**a**,**c**,**e**) and flesh (**b**,**d**,**f**) of tilapia during storage at 4 °C. (**a**,**b**) *B. subtilis*. (**c**,**d**) *Pseudomonas*. (**e**,**f**) Total number of bacteria. Different lowercase letters indicate significant differences at 0.05 level (*p* < 0.05).

**Figure 6 foods-10-03093-f006:**
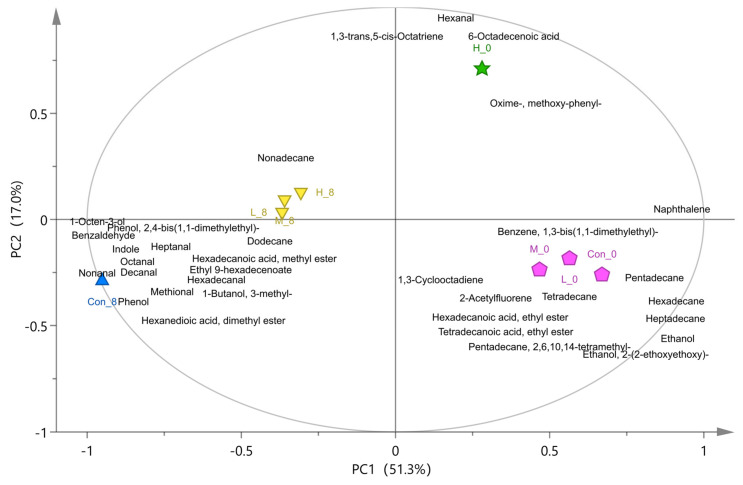
The PCA biplot of volatile compounds in different treatment groups. The control group (Con_0: Refrigerate for zero days; Con_8: Refrigerate for eight days) was fed with common feed, and the experimental groups (C6 + L (H_0, H_8), C15 + L (M_0, M_8), B18 + L (L_0, L_8)) were fed with the supplemented *B. subtilis* feed. Different colors represent different clustering groups.

**Figure 7 foods-10-03093-f007:**
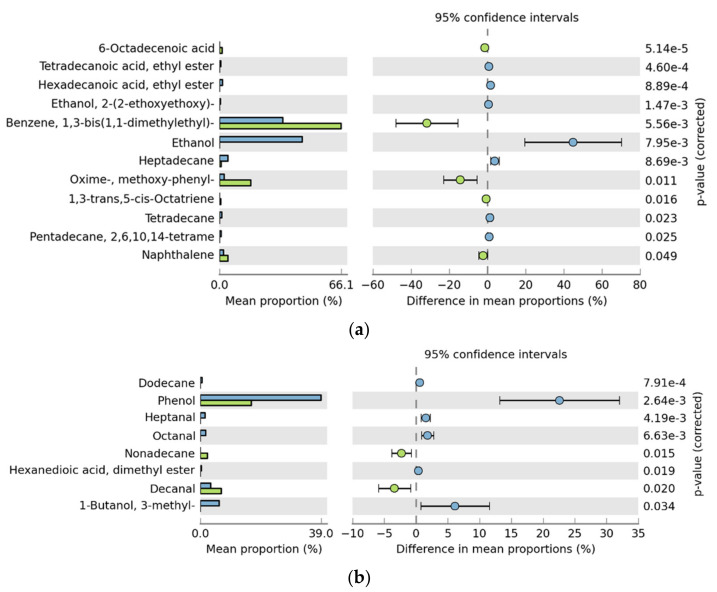
Difference analysis of control group (Con, blue) and high-adhesion *B. subtilis* C6 experimental group (H, green); (**a**) Refrigerated for zero days; (**b**) Refrigerated for eight days. (*p* < 0.05).

**Figure 8 foods-10-03093-f008:**
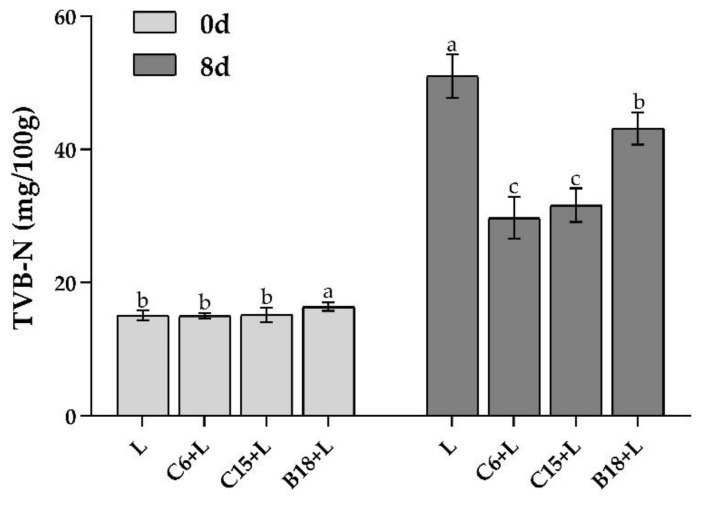
Changes in TVB-N content in fish flesh during storage. Different lowercase letters indicate significant differences at 0.05 level (*p* < 0.05).

**Figure 9 foods-10-03093-f009:**
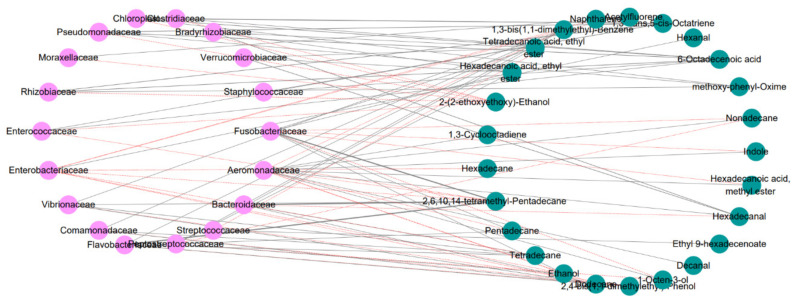
Correlation network analysis plot of intestinal flora and volatile flavor compounds. Negative correlation (red dotted line). Positive correlation (gray solid line).

**Table 1 foods-10-03093-t001:** The GenBank accession number of the nucleotide sequence of the experimental strain.

Strain	GenBank Accession Number
*Bacillus subtilis* B02	OL423511
*Bacillus subtilis* B08	OL423512
*Bacillus subtilis* B15	OL423514
*Bacillus subtilis* B18	OL423516
*Bacillus subtilis* C6	OL423515
*Bacillus subtilis* C15	OL423517
*Pseudomonas* spp. L.	OL423525

**Table 2 foods-10-03093-t002:** Specific primer sequence of the strain.

Strain	Primer	Sequence (5′→3′)
Total bacteria	27-F	AGAGTTTGATCCTGGCTCAG
	1429-R	GGTTACCTTGTTACGACTT
*Pseudomonas*	Pse-F	CTGCATCATGGCCGGTGACAACATTT
	Pse-R	GTCGCATGGCTGTCGGTCTTCAGATC
*Bacillus subtilis*	Bac-F	AAAGTCTGACGGAGCAACGC
	Bac-R	ACCGCCCTATTCGAACGGTA

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
