# Peer review of "The Application of Bacillus subtilis for Adhesion Inhibition of Pseudomonas and Preservation of Fresh Fish"

_foods, 2021, doi:10.3390/foods10123093_

Round 1

Reviewer 1 Report

Though the concept of the manuscript is worth the study, the manuscript required extensive rewrite with the support of a native language speaker before further processing.

The experimental procedures and detailing is not clear.

some of the comments

line 9: delete “of”

line 11, 12, 15, 34: Spp. is missing after the genus name. Check throughout the manuscript.

Line 13: expand the TVB-N

Line 13-17: incomplete sentence

Line 36: “,” missing

Line 39: italicize the bacteria name

Line 41: E. coli, use the full form when it appears first in the text

Line 41: ETEC?

Line 42: use abbreviation for Bacillus subtilis

Line 43-46: need clarity in the sentences

Line 49: what is TVB-N and K value?

Line 51, 52: italicize in vitro, in vivo. Do this in the rest of the manuscript also.

Line 59; sp.? is it spp.?

Line 64: rewrite the sentence

2.2. Determination of the Adhesion Ability of B. subtilis in Vitro: 2.4.1. Feeding Experiment of Tilapia: 2.5. Determination of Total Volatile Basic Nitrogen (TVB-N): the procedure is commending the readers like “add” “shake” “mix”

Line 80. What is “L”

2.3. Determination of the Adhesion Inhibition Ability of B. subtilis to Pseudomonas: rewrite

Line 145: The genomic DNA of fish intestine was amplified by PCR?

Reviewer 2 Report

My review report for the authors regarding the manuscript titled'Fresh-keeping Application of Bacillus subtilis based on
Adhesion Inhibition of Pseudomonas' is:

Major comments

1) The manuscript needs English language improvement e.g., syntax and grammar need improvement

2) The source of bacterial strains is mentioned; however, the taxonomic identity needs infomation. This means that accession number of the isolated strains should be mentioned

3) The whole M&M section is written in poor language. M&M should be written in passitive voice e.g., line 129 'Refer to Wang XF's method and make appropriate modification [24]'. Is this an order for the reader or information being the method used??

4) Discussion is poorly written. The results section is good and the authors produced a lot of data with a good presentation; however, should be justified by comparing with relevant studies and logical reasoning

Minor comments

1) Title is not clear and concise

2) Line 9: inhibiting the of spoilage...?

3) The objective of the study is not clear

4) The whole manuscript should be checked for typographical and structural errors. The MS is partly difficult to follow.

Round 2

Reviewer 1 Report

The author has improved the manuscript accordingly.

Author Response

Dear Editors and Reviewers:

Thank you for your letter and for the reviewers’ comments concerning our manuscript. We have studied comments carefully and have made correction which we hope meet with approval. Revised portion are marked in the revised format in the paper.

Reviewer 2 Report

The authors have significantly improved the MS in response to my comments

Author Response

(The authors gave the same response as above.)
